# Non-Criteria Manifestations of Juvenile Antiphospholipid Syndrome

**DOI:** 10.3390/jcm10061240

**Published:** 2021-03-17

**Authors:** Takako Miyamae, Tomohiro Kawabe

**Affiliations:** Pediatric Rheumatology, Institute of Rheumatology, Tokyo Women’s Medical University, Tokyo 162-8666, Japan; kawabe.tomohiro@twmu.ac.jp

**Keywords:** antiphospholipid antibodies, children, chorea, systemic lupus erythematosus

## Abstract

Antiphospholipid syndrome (APS) is a systemic autoimmune disorder mainly characterised by increased risks of thrombosis and pregnancy morbidity and persistent positive test results for antiphospholipid antibodies (aPLs). The criteria for diagnosing juvenile APS have yet to be validated, while the Sydney classification criteria do not contain several non-thrombotic clinical manifestations associated with the presence of aPLs. As such, difficulties have been encountered in the diagnosis of patients who have no certain thrombotic occlusions. Moreover, extra-criteria manifestations (i.e., clinical manifestations not listed in the classification criteria), including neurologic manifestations (chorea, myelitis and migraine), haematologic manifestations (thrombocytopenia and haemolytic anaemia), livedo reticularis, nephropathy and valvular heart disease have been reported, which suggests that the clinical spectrum of aPL-related manifestations extends beyond that indicated in the classification criteria. Studies have demonstrated that more than 40% of children with aPLs demonstrated non-thrombotic aPL-related clinical manifestations alone. Moreover, our results showed that the pathogenesis of non-criteria manifestations is characterised by “APS vasculopathy”. The present review introduces the characteristics and findings of non-criteria manifestations observed in juvenile APS.

## 1. Introduction

Antiphospholipid antibodies (aPLs) are antibodies directed against phospholipid-binding proteins, phospholipids and other proteins, with β2-glycoprotein I (β2-GPI), cardiolipin, prothrombin, annexin V, protein C and protein S being their most relevant targets. Antiphospholipid syndrome (APS) is a systemic autoimmune disorder mainly characterised by increased risks for thrombosis and pregnancy morbidity with persistent positive test results for aPLs [1]. The aPL detection tests included in APS classification criteria described later are anticardiolipin (aCL) antibody (IgG or IgM) enzyme-linked immunosorbent assay (ELISA), anti-β2-GPI antibody (IgG or IgM) ELISA and lupus anticoagulant (LA) assay [2]. The LA is characterized by correction of prolonged clotting time with added phospholipid but not with control plasma, which confirms that the coagulation inhibitor is phospholipid-dependent [3]**.**

APS had been described in detail during the 1980s, while the first papers had been published in 1983 [4]. With the discovery of aPLs and subsequent definition of APS, the effects on the coagulation pathway had initially been considered responsible for the occurrence of thromboses. However, the impact of aPLs on the endothelium and platelets had later become clear. Moreover, recent findings have shown that the actions of aPLs were multifactorial and could perhaps affect neural tissues and hormonal and complement pathways, consequently promoting non-thrombotic manifestations. Other aPLs were not included in the classification criteria. Of these aPLs, antibody to the phosphatidylserine–prothrombin complex (IgG or IgM) and IgA anti-β2-GPI antibody are the most critical antibody associated with the pathological manifestations caused by aPLs, not limited to the classification criteria [5].

In this review, we outline the limitations of the current APS classification criteria and non-criteria manifestations associated with aPLs in children.

## 2. Primary and Secondary Antiphospholipid Syndrome in Children

APS can occur by itself (i.e., primary APS) or in conjunction with other autoimmune conditions (i.e., secondary APS), with the latter being classically associated with systemic lupus erythematosus (SLE). The Ped-APS Registry, an international collaboration coordinated by the European Forum on Antiphospholipid Antibodies and Juvenile Systemic *Lupus Erythematosus* Working Group of the Paediatric Rheumatology European Society assembled between 2004 and 2007, revealed that 60 of 121 cases (49.5%) were considered primary APS, 60 (49.5%) were associated with a second autoimmune condition and one (1.0%) was associated with malignancy. Among the 60 cases associated with a second autoimmune condition, 46 cases of SLE (76.7%) and four of lupus-like disease (7.7%) had been observed [6]. Another study showed that low and non-pathogenic titres of aPLs could be detected in 1–5% of healthy people [7].

## 3. Thrombogenic Mechanisms in Antiphospholipid Syndrome

Thrombotic lesions are characterised by predominant non-inflammatory occlusive and/or mural thrombosis and subsequent consequences. Among various known aPLs, anticardiolipin (aCL), anti-β2-GPI and LA have had well-established clinical significance [3]. The continuous presence of aPLs, a hallmark of APS, is the most common cause of acquired thrombophilia associated with venous, arterial and small vessel thrombosis and pregnancy complications [8]. Multiple mechanisms can explain how aPLs catalyse clotting reactions, including direct interaction with proteins involved in initiating and controlling blood coagulation [9], such as anti-β2-GPI–β2GPI complexes, activated platelets, endothelial cells and monocytes. Endothelial cell activation induces proinflammatory cytokine production, causing increased leukocyte adhesion. Monocyte activation promotes the release of proinflammatory cytokines and tissue factors, stimulating coagulation, activation and ultimately fibrin production. In patients with APS, anti-β2-GPI–IgG can bind to circulating β2-GPI, creating stable complexes with platelet product factor 4 (PF4), a small chemokine belonging to the CXC chemokine family that is also known as chemokine (C-X-C motif) ligand 4 (CXCL4) [10]. This macromolecular structure (PF4-β2-GPI–anti-β2-GPI) significantly induces platelet p38MAPK phosphorylation, thromboxane B2 production and a conformational change in the GP IIb/IIIa receptor, all of which are markers of platelet activation [11]. These effects induce a procoagulant state, ultimately causing thrombosis, especially among those with complement activation and endothelial injury due to infection and trauma [12,13].

Hypocomplementemia can be observed in primary APS, showing serum levels of complement C3, C4 and CH50 [14]. In these case groups, hypocomplementemia was accompanied by a significant increase in serum C3a and C4a levels, which indicates that hypocomplementemia is caused by complement activation, which is thought to be due to activation of the classical pathway by immune complex formation. It has been interpreted that when aPLs induce complement activation, some type of biological stress exacerbates the complement activity and activates the C5a expression, which leads to thrombus formation. Cases of refractory APS and catastrophic APS that responded well to eculizumab, a monoclonal antibody directed against the complement protein C5, have been reported [15,16,17].

Thrombosis, therefore, represents the result of different pathogenic pathways with various impairments in clot formation, coagulation cascade, endothelial function and fibrinolysis occurring in APS.

## 4. Classification Criteria for Antiphospholipid Syndrome

At present, no diagnostic criteria have been established for APS yet. Diagnostic criteria are generally broad and should reflect the various disease heterogeneity features to accurately identify as many people with the condition as possible [18]. Conversely, the classification criteria are comprised of standardised definitions primarily intended to create well-defined, relatively homogenous cohorts for clinical research. Classification criteria have been established to select patients for inclusion in APS cohorts. One of the most prevalent classification criteria for APS requires the presence of at least one clinical event and one positive laboratory test for aPLs, including LA or aCL antibodies, in medium to high titres detected on two or more occasions at least 12 weeks apart. In 2006, the classification criteria were revised in Sydney, Australia [2]. European evidence-based recommendations for diagnosing and treating paediatric antiphospholipid syndrome by the single hub and access point for paediatric rheumatology in Europe (SHARE) initiative mention that non-thrombotic features are also frequently present in children in addition to thrombosis [19]. Some have even argued that children with clinical manifestations associated with aPLs only rarely fulfil the APS consensus criteria [20]. Childhood-onset APS has primarily been reported in patients with vascular thromboses and has been less frequently associated with isolated neurological or haematological manifestations [21,22,23]. The details will be more specific in a later section.

## 5. Non-Criteria Manifestations in Antiphospholipid Syndrome

The Sydney classification criteria do not contain several non-thrombotic clinical manifestations associated with the presence of aPLs. As such, diagnostic difficulties have been encountered in patients who exhibit no certain thrombotic occlusions. However, while many clinical manifestations can be explained by thrombotic occlusions of large vessels and predominantly small vessels, as in catastrophic APS [24,25], many others cannot. Therefore, the clinical spectrum of APS encompasses additional manifestations that can affect many organs and cannot be explained exclusively by a prothrombotic state (Table 1). Clinical manifestations not listed in the classification criteria (i.e., extra-criteria manifestations) include neurologic manifestations (chorea, myelitis and migraine), hematologic manifestations (thrombocytopenia and haemolytic anaemia), livedo reticularis, nephropathy and valvular heart disease. Besides, an analysis of paediatric cases of APS secondary to SLE revealed amaurosis fugax, osteonecrosis and interstitial pneumonitis, which is associated with the presence of aPL [26]. Indeed, studies have demonstrated that more than 40% of children with aPLs exhibited non-thrombotic aPL-related clinical manifestations alone [27]. Even if the presence of aPLs is confirmed and symptoms and findings are inferred to be related to aPLs according to the classification criteria but no thrombotic events have occurred, a diagnosis of APS cannot be made.

While some APS cases have evident thrombosis, others exhibit “APS vasculopathy” in the absence of thrombosis. In other conditions, including foetal loss, complement activation with neutrophilic tissue invasion occurs. Typical neurological manifestations of APS include ischaemic stroke and cerebral sinus vein thrombosis, both caused by thrombotic cerebral vessel occlusion [29,30]. Owing to arterial thrombosis, ischaemic stroke represents the most common neurological manifestation and the primary cause of morbidity and mortality in APS [31]. As such, the pathogenesis of non-criteria manifestations is characterised by “APS vasculopathy”. The following discussion summarises the non-criteria manifestations observed in juvenile APS.

### 5.1. Non-Criteria Neurological Manifestations in Antiphospholipid Syndrome

Some neurological manifestations have been associated with the presence of aPLs, including various movement disorders, epilepsy, migraine, cognitive defects, psychiatric diseases, transverse myelitis, multiple sclerosis-like disorders, sensorineural hearing loss and Guillain–Barré syndrome [32,33]. However, the thrombogenic effects of aPLs cannot fully explain these manifestations, resulting from both thrombotic and non-thrombotic immune-mediated mechanisms such as direct interactions between aPLs and neuronal tissues or immune complex deposition in the cerebral blood vessels wall [34,35,36]. In children, the most common non-stroke neurological manifestations observed in APS are migraine headaches (7%), chorea (4%) and seizures (3%) [20], with only migraines (20.2%) and epilepsy (7%) being prevalent among the adult Euro-Phospholipid cohort [31]. A prospective study of non-thrombotic neurological manifestations in 333 adult patients with APS reported that chorea, migraines and epilepsy occurred more often in secondary APS. By contrast, headaches and depression occurred more frequently in primary APS. Moreover, significant associations were found between the presence of aCL IgG and acute ischemic encephalopathy in secondary APS, aCL IgM and epilepsy in secondary APS, aCL IgM and migraine in primary APS, β2-GPI IgG and chorea in secondary APS and β2-GPI IgM and transient ischemic attack and epilepsy in primary APS. They also showed that LA was linked to depression, transient global amnesia and migraine in those with primary APS [37].

#### 5.1.1. Migraine

Migraine, the most commonly reported type of headache in APS/aPL-positive patients, is characterised by severe, mostly unilateral excruciating pain associated with photophobia, phonophobia, nausea and vomiting. One meta-analysis reported a significant association between aPLs and migraine in adults [38], although no paediatric data have been available. Thrombotic and platelet dysfunctions have been hypothesised to clarify the pathogenic role of aPLs in the development of headaches [39]. Some studies have reported platelet dysfunction during migraine attacks and that the binding of aPLs to platelet membrane phospholipids causes platelet activation [40,41,42,43]. Hence, further elucidation of the pathophysiology of aPL-associated migraine, especially concerning platelets and vascular endothelial cells, is needed.

#### 5.1.2. Chorea

Chorea is defined as a hyperkinetic movement disorder characterised by excessive spontaneous movements resulting from basal ganglia damage due to various causes in children. Disorders presenting chorea include infections, ischaemia during cardiopulmonary bypass, “extrapyramidal” cerebral palsy, different toxic and acute metabolic processes, degenerative conditions and inborn errors in metabolism. The association between chorea and the presence of LA had first been described in 1983 [44]. A study from the European Phospholipid Project Group examining a cohort of 1000 patients with APS, including both primary and secondary APS, demonstrated that chorea developed in only 13 patients [45]. aPL-related chorea has more frequently been observed in children and females.

Fundamentally, neuroimaging studies have rarely demonstrated ischaemic basal ganglia lesions in patients with antiphospholipid antibody-related chorea. Moreover, the clinical benefits of corticosteroids and other immunosuppressants against chorea manifestations strongly support an autoimmune mechanism rather than a vascular one [46]. In the neurological involvement of APS, aPLs bind to the brain endothelium, causing endothelial dysfunction that leads to micro-thrombosis and blood vessel inflammation [47,48] (Figure 1). This binding is thought to disrupt or alter the permeability of the blood–brain barrier, a highly selective semi-permeable barrier consisting of an endothelium that prevents circulating blood from non-selectively crossing into the extracellular fluid of the central nervous system, as well as extravasation of neurotoxic cytokines and serum proteins, including aPLs and activated thrombin [49,50]. Subsequently, aPLs cause direct neural damage by binding to the neural cell surface. Reports of studies that used in vitro experimental APS models have shown that aPLs bind to neural and astrocyte membranes [51,52,53]. The binding of aPLs to the phospholipid-rich areas in the basal ganglia, directly depolarizing the neurons and causing damage, has been considered the mechanism by which aPL-related chorea occurs [54].

Three retrospective and demographic analyses of aPL-related chorea [55,56,57] demonstrated that most patients were female with unilateral and bilateral chorea. The most recent study with the largest cohorts investigating the long-term outcome in 32 patients [57] found that chorea associated with SLE or aPLs occurred early during connective-tissue disease and showed good outcomes. However, relapse was observed in eight (25%) patients during follow-up (12.2 ± 11.3 years). Moreover, 12 patients (37.5%) developed arterial thrombosis, indicating the substantial risk for developing thrombosis among these patients. The authors concluded that prophylactic treatment with antithrombotic therapy might reduce the risk of further thrombosis (8 vs. 57%; *p* = 0.01). Patients who exhibited aPL-related chorea as the initial symptom did not fulfil the Sydney criteria for APS upon onset.

### 5.2. Non-Criteria Haematologic Manifestations of Antiphospholipid Syndrome

Haematologic disorders, including immune thrombocytopenia, autoimmune haemolytic anaemia and Evans syndrome, have been the most frequent non-criteria manifestations of juvenile APS, occurring in 30–50% of the cases [6].

Thrombocytopenia is the most relevant non-criteria manifestation of APS, with a reported prevalence of 20–50% of adult cases [55]. Moreover, it had been observed in 20–25% of children with APS, often in association with Coombs-positive haemolytic anaemia and Evans syndrome [6,58]. A case-control study of 42 children diagnosed with immune thrombocytopenic purpura detected IgG aCLs in 78% and anti-β2GPIs in all chronic cases [59].

The pathogenesis of thrombocytopenia related to aPL antibodies has been attributed to increased platelet destruction or abnormal platelet pooling [60]. The increased platelet destruction is possibly caused by the direct binding of anti-β2-GPI antibodies or anti-β2-GPI–β2-GPI complexes. Anti-β2-GPI-IgG can bind to circulating β2-GPI, creating stable complexes with PF4, the dominant β2-GPI binding protein [10,11]. Human platelets secrete PF4 tetramers, allowing recognition and enhancing anti-β2-GPI–β2-GPI interactions in APS [61]. In the presence of β2-GPI and anti-β2-GPI antibodies, the activation-signalling pathway of platelets is mediated by platelet membrane receptors, apolipoprotein E (ApoER2) and glycoprotein Ibα (GPIbα). This interaction significantly induced platelet P38MAPK phosphorylation, thromboxane B2 production and P-selectin and glycoprotein IIb/IIIa expression [11,62]. Platelet activation induced by aPLs and the destruction of platelets by antibodies directed against their membrane glycoproteins increased by aPLs, have been the primary immune mechanisms causing lower platelet count in APS [63]. Antiplatelet autoantibodies that trigger complement activation via the classical complement pathway promote complement-mediated platelet destruction [64] (Figure 2).

### 5.3. Non-Criteria Cardiovascular Manifestations in Antiphospholipid Syndrome

Heart valve disease (HVD) in patients with aPLs is defined as valve lesions and/or moderate to severe valve dysfunction detected through transthoracic or transoesophageal echocardiographic/doppler studies [65]. Libman and Sacks had described HVD based on the presence of valve thickening or vegetation (mainly mitral and aortic) in patients with SLE. It has frequently been detected in adult patients with APS, with a prevalence reaching as high as 65% [66]. However, HVD has not been extensively investigated in juvenile patients [67]. Although the mechanisms through which HVD develops remain unclear, aPLs have been suspected of playing a substantial role in its pathogenesis [68].

### 5.4. Non-Criteria Cutaneous Manifestations in Antiphospholipid Syndrome

The Ped-APS Registry also reported cutaneous disorders in 18% of the cases. Accordingly, the most common cutaneous manifestations reported in children included in the paediatric APS registry have been livedo reticularis (LR, 6%), Raynaud phenomenon (6%) and skin ulcers (3%) [6]. Blood flow disturbances cause LR in small- and medium-sized blood vessels within the dermal–subdermal junction [68]. LR histopathology shows endothelitis and obliterating endarteritis with no evidence of true vasculitis consistent with “APS vasculopathy”.

### 5.5. Non-Criteria Renal Manifestations in Antiphospholipid Syndrome

The kidneys are significant targets for damage in APS. Clinical renal manifestations have been defined based on the site and size of the vessels involved [69]. All vessels, veins and arteries (from renal arteries to glomerular tuft capillaries) can be engaged in the renal syndromes in APS [70]. The renal lesions in APS related to thrombosis include renal artery occlusion or infarction and resulting hypertension and renal vein embolism. Moreover, non-criteria renal involvement includes intrarenal vasculopathy, APS nephropathy (APSN) and glomerular disease (Table 1).

## 6. Mechanisms Underlying Non-Criteria Manifestations Associated with aPLs and Potential Targeted Therapies

The pathogenesis of APS is not yet fully understood; however, several mechanisms have been reported in recent years. The mechanistic target of rapamycin complex (mTORC) pathway has been identified to be activated in the vascular endothelium of proliferating intrarenal vessels and a putative therapeutic target in vascular lesions associated with APSN [71]. Moreover, complement activation, which is also a mechanism of thrombus formation as mentioned earlier, has been reported to be effective for TMA caused by SLE and APS with eculizumab, which suggests its association with non-criteria manifestation [72]. Furthermore, several findings indicate that APS is characterised by increased oxidative stress. Oxidative stress and mitochondrial dysfunction are involved in the pathophysiology of atherothrombosis in APS and SLE. aPLs induce nitric oxide and superoxide production, which results in increased levels of plasma peroxynitrites [73]. Inhibition of oxidative stress could contribute to molecular and cellular targets for the treatments of APS and SLE. Dabigatran, a thrombin inhibitor, and statins reduce vascular oxidative stress and inflammation and improve endothelial function [74].

## 7. Conclusions

Although a cohort of homogeneous patients is necessary to establish, classification criteria are of only relative usefulness in clinical diagnosis. A study on patients with clinical features not currently included in the classification criteria for diseases would allow for the analysis of different stages and forms of the disease as well as improve our understanding of the pathogenesis. It could reduce the risk of underdiagnosis/undertreatment of the condition. Furthermore, early detection of aPL-related complications could institute prompt management. In the future, it is desirable to establish guidelines for the diagnosis and treatment of “aPL-related syndromes”, including the non-criteria manifestations.

## Figures and Tables

**Figure 1 jcm-10-01240-f001:**
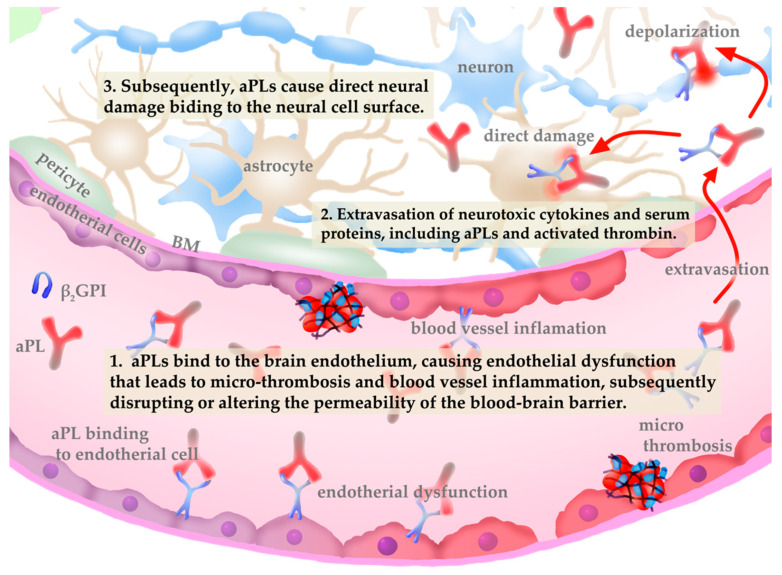
Pathophysiology of central nervous system vasculopathy induced by antiphospholipid antibodies (aPLs).

**Figure 2 jcm-10-01240-f002:**
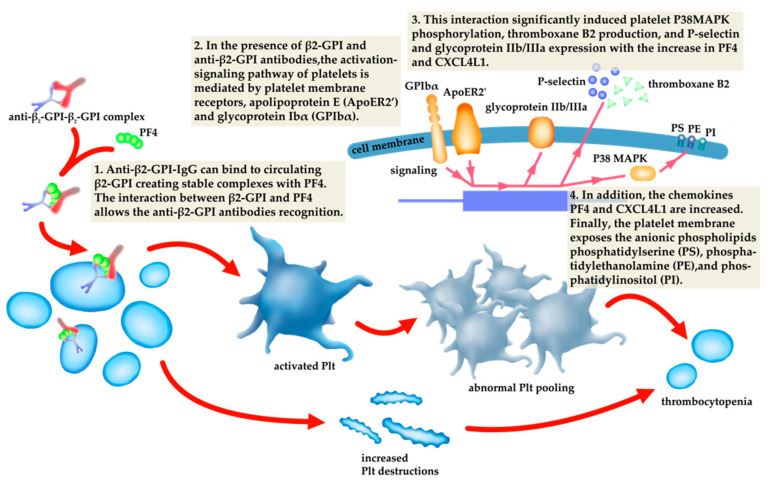
The relationship of the pathogenesis of thrombocytopenia to antiphospholipid antibodies (aPLs).

**Table 1 jcm-10-01240-t001:** Main non-criteria manifestations of antiphospholipid syndrome reported in children manifestations described in this paper.

Neurologic	Ophtalmoogic	Hematologic	Dermatologic	Musculosceletal	Cardiac	Pulmonary	Renal	Endocrine
Chorea *	Amaurosis Fugax [26]	Thrombocytopenia *	Livedo reticularis *	Osteonecrosis [26]	Valvular disease *	Pulmonary Hypertension	Intrarenal vasculopathy; APS neprhopathy (APSN) *	Adrenal insufficiency
Migraine/Headache		Hemolytic anemia	Raynaud’s Phenomenon	Arthritis [28]		Intestinal pneumonitis [26]	Glomerular disease	
Seisure/Epilepsy		Evans syndrome	Purpura fulminans				Hypertension	
Transverse myelitis		Leukopenia	Skin ulcers					
Pseudotumor cerebri		Bleeding diathesis	Pseudocasculitis lesions					
Mood disorder			Chronic urticaria					
Cognitive impairment								

* manifestations described in this paper.

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
