# Peer review of "Non-Criteria Manifestations of Juvenile Antiphospholipid Syndrome"

_jcm, 2021, doi:10.3390/jcm10061240_

Round 1

Reviewer 1 Report

This is a brief but well-focused review of the characteristics of APSin children, although there are some confusing aspects that must be modified.

Introduction: The description of the aPLs should be done in two stages. In the first paragraph, only the aPLs included in the classification criteria (anti B2GP1 and anti cardiolipin of IgG and IgM isotypes) should be described. In another paragraph and precisely because this article deals with the shortcomings of the current classification criteria, it should be mentioned that there are also gaps in the laboratory criteria and the extra-criteria antibodies should be named. This list of extra criteria-aPL should include the two most associated with the symptoms of APS: antibodies against phosphatidylserine / prothrombin complex (IgG / IgM) and anti-B2GP1 of the IgA isotype.

Table 1: APS in children was also associated with amaurosis fugax, osteonecrosis and interstitial pneumonitis (PMID: 14667097).

Line 77: in this sentence, when saying that there are no criteria for children, it seems to imply that there are for adults, when they don't exist. The statement must be for ALL APS, not just limited to children. Lines 77-80 It gives the impression that the concepts of diagnostic criteria vs. classification criteria are being assimilated as synonyms. The authors affirm that in order to diagnose PSA, at least one clinical (classification) criterion and one laboratory criterion are necessary, that is, the current classification criteria. However, later on, the authors explain the differences by citing greater rigor of the classification criteria (line 84) and make it clear that diagnostic criteria and classification criteria are not the same when citing PMID: 25776731. The diagnosis of autoimmune diseases, such as APS or SLE, continues to be a clinical process in which the signs and symptoms included in the classification criteria can be used as tools, but in which the experience and sagacity of the physician are essential.

Authors should use more descriptive phrases in relation to this situation, something like:

“At the present time, there are still no diagnostic criteria for APS. However, there are established classification criteria for the purpose of selecting patients to be incorporated into APS cohorts but not intended to capture the entire universe ..... ".  (It is a suggestion for the authors to understand that in the manuscript there is a problem of adequate expression of the situation).

Lines 257-262 The list of kidney disorders related to antiphospholipid antibodies has been made in table format but without considering it as such. It should be incorporated into table 1, included as common text within the paragraph, or converted into table 2.

The conclusions should be rewritten to avoid confusion of the concepts of diagnosis and classification. The classification criteria, although necessary to establish homogeneous cohorts of patients, have only a relative utility in the clinical diagnosis. The study of patients with clinical characteristics not currently included in the classification criteria will allow studying the different stages and forms of the disease, which will help to better understand the pathogenesis.

Reviewer 2 Report

I read with great interest the review “Non-criteria manifestations of juvenile antiphospholipid syndrome”.

The authors analyze that the pathogenesis of non-criteria manifestations in pediatric antiphospholipid syndrome emphasizing that a significant proportion of affected children demonstrate non-thrombotic aPL-related clinical manifestations alone.  In particular they elucidate the related mechanisms of not thrombotic manifestations that could be explained as “APS vasculopathy”. Currently, there are no accepted validated criteria for pediatric APS. To improve the awareness of the many patterns of APS presentations in children could reduce the risk of underdiagnosing/undertreating this condition. Furthermore, early detection of the APS-related complications could institute a prompt management.

For these reasons the purpose of the manuscript is meaningful in pediatric rheumatology set. But I strongly suggest to add some sentences at the end of introduction about the aim of this extensive review paper. At least it is going to give some idea to reader what the purpose of this review.

The review is clearly written in general and well organized. The figures are very informative.

However, I suggest:

  • In the “Classification Criteria for Antiphospholipid Syndrome”, line 80, please revise as a 12 week period instead of a 6-week period
  • Groot et al paper is one of the most important paper about the APS in childhood, but unfortunately authors did not mentioned and cite. In page 2, subheading 4, it is the best place to discuss this article.

Groot N, de Graeff N, Avcin T, Bader-Meunier B, Dolezalova P, Feldman B, Kenet G, Koné-Paut I, Lahdenne P, Marks SD, McCann L, Pilkington CA, Ravelli A, van Royen-Kerkhof A, Uziel Y, Vastert SJ, Wulffraat NM, Ozen S, Brogan P, Kamphuis S, Beresford MW. European evidence-based recommendations for diagnosis and treatment of paediatric antiphospholipid syndrome: the SHARE initiative. Ann Rheum Dis. 2017 Oct;76(10):1637-1641. doi: 10.1136/annrheumdis-2016-211001. Epub 2017 May 4. PMID: 28473426.

  • In the table 1, I kindly suggest you to add Articular involvement (Arthritis) (see Ma J et al, Clin Rheumatol. 2018).
  • The paragraph 5.5.1 “Antiphospholipid syndrome-associated nephropathy (APSN)” should be delated, because some sentences repeated at the conclusion
  • It might be better to write a short paragraph about the treatment date of the non-criteria manifestations.
  • The conclusion paragraph is too short, you should emphasize better the unmet needs about the topic. I also suggest to add future research agenda.

Reviewer 3 Report

The authors do not make a compelling case for the supposed non-criteria manifestations of APS. Many, if not all, of the clinical and laboratory manifestations they discussed and listed on Table 1 are potential manifestations of SLE or closely related conditions, such as MCTD, irrespective of the aPL antibody status, though a positive aPL antibody would classify the APS as secondary. To make a more compelling case, they need to add another 1 or 2 sections with several more references dedicated to more in-depth discussion of primary APS highlighting the uniqueness of the supposed non-criteria manifestations in the context of primary APS as the underlying etiology.

Minor comments:

Lines 35-36: The phrase “in the effects” is repetitive and should deleted

Lines 41-42: The phrase “autoimmune conditions” should be singular

Lines 73-74: The word “variously” should be changed to “various”

Line 80: Diagnostic timeframe is 12 weeks apart, not 6 weeks

Line 132: The lone word sentence “APS” should be deleted

Line 177: Inappropriate space in the word “inflammation” should be corrected

Line 190: Inappropriate use of parenthesis closure “)” (after the word “risk”) should be deleted

Line 198: The sub-subsection 5.2.1 is unnecessary and should be deleted; a mention of thrombocytopenia referring to Figure 2 should instead be added under subsection 5.2

Figures 1 and 2: Recommend shrinking the figures to 1/2 or 2/3rd their current size; also need to add numbers on the figures to match the figure legend descriptions

Lines 263-267: The sub-subsection 5.5.1 is unnecessary and should be deleted; the paragraph under this sub-subsection should be deleted as it is verbatim of the paragraph under conclusions

Lines 268-272: The heading says “conclusions” (plural), yet the authors wrote only one sentence; they need more elaboration on the concluding remarks (ideally should be 3-4 times the current length)

Round 2

Reviewer 3 Report

The authors have responded appropriately to my suggestions. 

Author Response

In response to your suggestion, we have conducted another native English editing.